# communications
# engineering

# Unsupervised learning of pixel clustering in Mueller matrix images for mapping microstructural features in pathological tissues

Jiachen Wan[1,9], Yang Dong[1,9], Yue Yao[1], Weijin Xiao[2], Ruqi Huang [3], Jing-Hao Xue[4], Ran Peng[5], Haojie Pei[1], Xuewu Tian[6], Ran Liao[1], Honghui He [1], Nan Zeng[1], Chao Li[2,7] & Hui Ma [1,8✉]

In histopathology, doctors identify diseases by characterizing abnormal cells and their spatial organization within tissues. Polarization microscopy and supervised learning have been proved as an effective tool for extracting polarization parameters to highlight pathological features. Here, we present an alternative approach based on unsupervised learning to group polarization-pixels into clusters, which correspond to distinct pathological structures. For pathological samples from different patients, it is confirmed that such unsupervised learning technique can decompose the histological structures into a stable basis of characteristic microstructural clusters, some of which correspond to distinctive pathological features for clinical diagnosis. Using hepatocellular carcinoma (HCC) and intrahepatic cholangiocarcinoma (ICC) samples, we demonstrate how the proposed framework can be utilized for segmentation of histological image, visualization of microstructure composition associated with lesion, and identification of polarization-based microstructure markers that correlates with specific pathology variation. This technique is capable of unraveling invisible microstructures in non-polarization images, and turn them into visible polarization features to pathologists and researchers.

[1] Guangdong Engineering Center of Polarization Imaging and Sensing, Tsinghua Shenzhen International Graduate School, Tsinghua University, Shenzhen 518055, China. [2] Department of Pathology, Clinical Oncology School of Fujian Medical University, Fujian Cancer Hospital, Fuzhou 350014, China. [3] Tsinghua-Berkeley Shenzhen Institute, Tsinghua University, Shenzhen University Town, Shenzhen 518071, China. [4] Department of Statistical Science, University College London, London WC1E 6BT, UK. [5] The School of Basic Medical Sciences, Fujian Medical University, Fuzhou 350122, China. [6] Department of Pathology, University of Chinese Academy of Sciences Shenzhen Hospital, Shenzhen 518106, China. [7] Fujian Key Laboratory of Translational Cancer Medicine, Fuzhou 350014, China. [8] Department of Physics, Tsinghua University, Beijing 100084, China. [9] These authors contributed equally: Jiachen Wan, Yang Dong.
✉email: mahui@tsinghua.edu.cn

In histopathology, doctors study the abnormality of tissue microstructure caused by diseases[1]. For instance, cancers are closely related to various microstructure alterations, such as changes in cellular nucleus size and anisotropy[1], collagen organization[2], or even composition of organelles in cytoplasm[3,4]. As the gold standard in pathology, hematoxylin and eosin (H&E) staining reveals tissue morphology and microstructure by dyeing the nucleus and extracellular matrix with varying colors of blue and pink[5]. Since malignant lesions induce changes in tissue microstructures, cancer-type-specific microstructural patterns observed on H&E slides provide crucial diagnostic information.

It has been known that interactions between polarized photons and complex media encode microstructural information. The $4 \times 4$ polarization transformation matrix, i.e., Mueller matrix, provide a comprehensive representation on the specimen's polarimetric properties, while non-polarization or other polarization methods only reveal a subset of them. Since polarization properties are closely related to the microstructural features of scattering media[6,7], particularly sensitive to super resolution features at subwavelength scale[8,9], Mueller matrix microscopy is capable of mapping tissue architecture down to subcellular level[9,10]. Being treated as an inverse problem, the pixel-level correlation between microstructure and polarization features are usually established by taking supervised learning approaches[11–14], as illustrated in Fig. 1. Pathologist segments regions on the H&E-stained pathological images to provide spatial labeling of the interested pathological structures, which in turn induces labeled pixels on the co-registered polarization images as the ground truth. Then polarization features corresponding to the labeled pixels are identified using supervised machine learning techniques. Such analysis can be performed on a low-magnification optical system, based on the finding that polarimetric imaging's contrast mechanism is insensitive to image resolution[15]. This approach has been used to study a variety of biological tissue samples through the extraction of microstructure-specific polarization features[11,14,16,17]. In particular, microstructures in liver tissue during lesion have been analyzed to extract the simplest form of polarization feature parameter for cancerous cell nucleus identification in hepatocellular carcinoma samples[16].

A Mueller matrix image encodes abundant microstructural patterns that potentially correlate with pathological variation. In this paper, we advocate an unsupervised learning approach to clustering pixels based on distinctive polarization features, which allows for identifying specific spatial organization via projection from clustered pixels to H&E images. Such unsupervised learning approaches provide advantages for experienced pathologists to exploit the rich information in Mueller matrix images and their medical expertise to identify pathological features which may not be clearly seen under normal optical microscopes in clinics.

In this study, we take full advantage of the microstructural information encoded in Mueller matrix images by adopting the unsupervised approach, as illustrated in Fig. 1. We cluster the polarization pixels and discover that these polarization clusters correspond to distinct pathological structures. Such correspondence between clusters and pathological structures appears resilient to individual differences of patients. It indicates that by combining Mueller matrix imaging with unsupervised pixel-level clustering, we can effectively decompose the histological structure into its basic constituent microstructure subtypes. We can further extract the microstructure subtypes that are sensitive to pathological variations, identifying them as polarization markers for assisted diagnostic purposes. We demonstrate the viability of the proposed method by applying it to analyze the H&E-stained pathological slides of liver cancer, which has been examined previously using Mueller polarimetry and supervised learning[16], and show that the identified clusters in polarization feature space segment a variety of pathological structures. We propose a set of promising polarization markers that can distinguish intrahepatic cholangiocarcinoma (ICC) from hepatocellular carcinoma (HCC), recognize HCC lesion, and identify HCC with different cellular differentiation degrees. This framework of dividing tissues into microstructural subtypes will be referred to as the tissue microstructural composition analysis in this article. The framework allows the exploration of the super-resolution microstructures, which are invisible in the non-polarization optical images for routine clinical diagnosis. Optimistically, it may assist pathological structure identification and quantitative diagnostic evaluation, and become a new tool for pathological research and practice.

## Method

**Samples.** HCC accounts for 70% of primary liver tumor, one of the cancers with the worst survival rate (20%)[18]. Undoubtedly, accurate staging of HCC subtypes has impactful clinical implications. As the second most common type of primary liver tumor, ICC is much more aggressive than HCC, and the differentiation between them remains challenging[19–22]. Here, we use HCC and ICC samples to demonstrate the validity of our proposed method. Pathological slides of all patients are provided by Fujian Medical University Cancer Hospital. Sampled from 41 patients, a total of 222 ROIs are used for the unsupervised model construction, in which 181 ROIs are labeled. Among them, 71 ROIs are from ICC samples, 85 ROIs are from HCC samples, and 25 ROIs are from normal tissues around lesions. The thickness of pathological slides is around 4 microns, which is within the single scattering regime and usually corresponds to small depolarization. The study was approved by the Ethics Committee of Fujian Medical University Cancer Hospital (SQ-2022-103-01), and participants provided written informed consent to take part in the study.

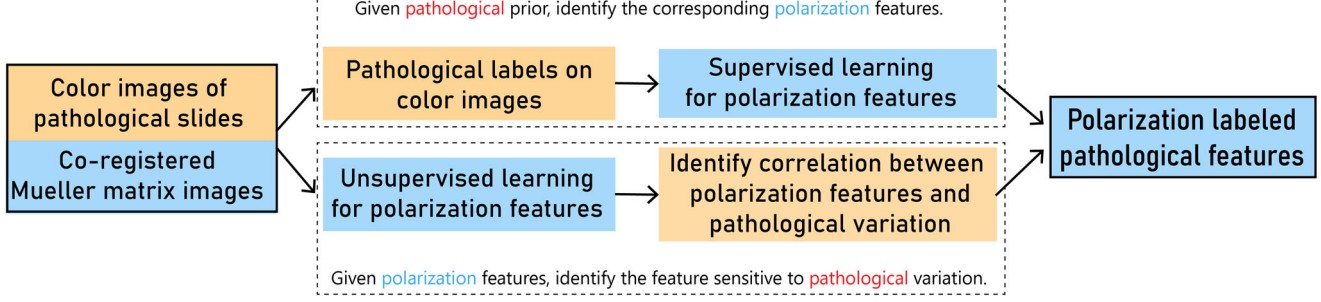

**Fig. 1 Two different frameworks to identify the correlation between polarization and pathological features.** The top path corresponds to the supervised learning approach, where polarization features are extracted for the pathological labels. The bottom path corresponds to the unsupervised learning approach, where the polarization pixels are first clustered into subtypes, and the subtypes that correlate with pathological variation are identified.

**Mueller matrix images**. Histological slides obtained from biopsy are measured using Mueller matrix microscope under $20 \times 0.4$NA objective lens with 633 nm wavelength to obtain the complete Mueller matrix ($4 \times 4$) images. The choice of this moderate NA allows us to achieve spatial resolution of 1 micron for resolving subcellular structures but maintain optimal accuracy in the Mueller matrix measurements. The homemade Mueller matrix microscopes were upgraded from low-cost commercial upright transmission microscopes by adding polarization optics modules in the existing optical path. Most of the samples were imaged by a MMM with a pair of fixed polarizer and rotating waveplate for both the polarization state generator and analyzer[10,23], and a $1001 \times 1301$ pixels 12-bit CCD camera (74-0107 A, QImaging, Canada). It is capable of taking a Mueller matrix image in about 3 min with less than 0.02 average root mean square error (RMSE) using standard samples including air, linear polarizer, and waveplate. Details for optimization and calibration of the microscope can be found in previous studies[10,23–25]. A small number of samples were imaged by an upgraded MMM[26] whose polarization state analyzer and 2D detector were replaced by two $2048 \times 2448$ pixels 16-bit division of focal plane (DoFP) polarimeters (PHX050S-PC, Lucid Vision Labs Inc., Canada). Extinction coefficient of the DoFP is 150 at 633 nm wavelength. The new MMM is capable of more than a factor of 10 increase in speed and a factor of 2 improvement in average RMSE, compared with rotating waveplate MMM. Detailed specifications of both MMMs can be found in reference[26]. RMSE can be reduced further by rotating both the polarizer and the wave plate in the polarization state generator[27].

Once the Mueller matrix images are obtained, a Gaussian filter is applied to reduce pixel value fluctuations while preserving the polarization features of microstructures, because polarization features is observed to be insensitive to changes in image resolution[15]. Through the sum decomposition of Mueller matrix introduced by Cloude, the physically unrealizable part of Mueller matrices is filtered[28].

**Polarization super-pixel**. Super-pixel approach is a proven method to reduce data redundancy while decreasing computational complexity for downstream tasks in the field of image processing[29–32]. Considering the large number of polarization pixels to be processed, a polarization super-pixel strategy is taken in our scheme to compress the volume of polarimetric data while preserving its main polarization features[13].

Specifically, the polarization super-pixels are created with the following steps in this work. First of all, we standardize the Mueller matrix elements of individual polarization pixels by removing the mean and scaling them to unit variance[33]. Then, we apply the minibatch KMeans algorithm[34] to all the polarization pixels in the ROI and group them into 1024 clusters, using the 15 Mueller matrix elements normalized by M11 as features. Lastly, the average of the polarization basis parameters[17] is calculated and recorded for each polarization super-pixel. Note that we can trace back the coordinates of the pixels that belong to the specific super-pixel.

Instead of preserving the entirety of local information by considering all the pixels, we characterize the local subset of neighboring points in polarization feature space with their centroids, and perform unsupervised algorithms on the set of centroids. Note that the spatial constraint of generic super-pixels is waived since it increases the compression rate for polarization data. Using the described super-pixel approach, pixels with similar polarization characterization are grouped, creating N polarization super-pixels for each ROI. N can be chosen based on the complexity of the images in polarization space. A larger N preserves more polarization information, while a smaller N leads to more compact data representation. Effectively, computing at the scale of super-pixel level instead of pixel level allows us to ignore the detailed local structure of polarization data and focus on the global structure in polarization space, improving robustness of the result. Through the aggressive use of super-pixels, the data volume is decreased by 3 orders of magnitude.

**Polarimetric basis parameters**. Mueller matrix describes comprehensively the sample's polarization properties, which provides extensive microstructural information. However, the explicit relationships between Mueller matrix elements and the characteristic microstructures are usually unclear, and many of the elements are sensitive to sample orientation. To accommodate this shortcoming, various decomposition[35–37] and transformation[38–41] methods have been proposed to provide polarization parameters which are functions of Mueller matrix elements but tend to have more explicit interpretability from the perspective of physics. These polarization parameters are used as the polarization basis parameters (PBP) for further statistical analysis[11,14,17]. A comprehensive list of the used PBPs is provided in Supplementary Table 1. More details are also available from reference[17,38].

**UMAP and clustering**. Polarimetric data lie on a nonlinear high-dimensional manifold, so we used uniform manifold approximation and projection (UMAP) for analysis, which offers non-linear dimensionality reduction and projection of unseen data[42]. UMAP reveals the underlining structure in the polarization data, and the clusters in the UMAP space may correspond to specific pathological structures, which are annotated and studied. UMAP is used to visualize the global structure of polarization features in this paper. After normalizing the super-pixels into zero mean and unit variance, they are fed into the UMAP model using Canberra distance and 30 neighbors. We chose the Canberra distance since it is robust to outliers and sensitive to data around the origin. With polarimetric data, Canberra distance produces superior clustering results than conventional metrics such as Euclidean and Manhattan distance, a phenomenon observed in other data types as well[29,43,44]. In UMAP model, the number of neighbors parameter controls the size of local neighborhood during the manifold approximation process. Decreasing the parameter means preserving more local information and shortening the computation time. Empirically, number of neighbors is set as 30, so it is near the sweet spot for both global structure preservation and optimal computation time.

Once the data is projected onto the two-dimensional UMAP axes, hierarchal clustering is applied[45]. Single linkage is used, where the distance between two clusters is determined by the pair of points that are the closest to each other, one from each cluster. To focus on the main structure of the data, the points with low surrounding density are removed before clustering, by using a $200 \times 200$ regular grid, and using 11 as the cut-off threshold to remove the low-density areas. The cluster labels of the removed points are recovered using a semi-supervised approach, determined by label spreading after clustering is conducted using the trimmed data points[46]. In particular, a 41-fold cross-validation is conducted, where ROIs from 1 patient is reserved as unseen test data while the rest is used to construct the UMAP model. Especially, in each fold, we compute the Euclidean distance between the projection onto the UMAP axis of the test ROIs and its projection onto the UMAP axis established with the whole data from 41 patients. The mean distance is 0.012 with variance 0.00068, normalized with respect to the range of $x$- and $y$-axis,

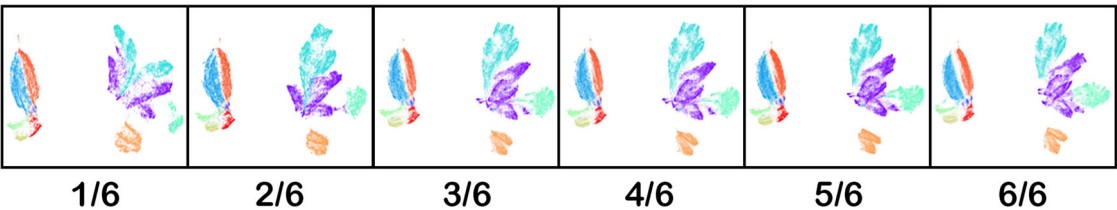

**Fig. 2 Convergence of the overall configuration in UMAP space as the number of ROIs increases.** The numbers indicate the fraction of data used, increasing at a step of 1/6, equivalent to 37 regions of interest (ROI) from samples of 6–7 patients. Above the numbers are the corresponding Uniform Manifold Approximation and Projection (UMAP) using the specified fraction of total data.

suggesting a stable UMAP configuration with respect to the training set.

To verify the plausibility of decomposing tissue structures into a stable basis of microstructural clusters using polarization imaging, we first analyzed 37 ROIs containing characteristic lesion structures, and closely monitored the changes in UMAP projection as we added more ROIs from new patients covering a variety of pathological structures. Using as little as 5% of the total dataset, we observe clusters forming in the UMAP projection. As we gradually increase the number of patients used during UMAP training process, the overall configuration converges, as shown in Fig. 2. Different structures require different amount of data to converge. For instance, the configuration of the two clusters on the left (cluster 1 and 2) starts to emerge with as few as two patients, and remains stable. Other clusters, especially the two clusters on the right (cluster 3 and 4), struggle during convergence due to their complex microstructure characteristic. Overall, we confirm that the macroscopic structure of the UMAP projection converges after roughly 20 patients (or equivalently 100 ROIs), and the layout remains consistent as the data size increases even further.

**Reporting summary**. Further information on research design is available in the Nature Portfolio Reporting Summary linked to this article.

## Results

**Polarimetric feature-based microstructural mapping**. Since Mueller matrix describes how the optical and structural properties affect polarization states as the photons propagate and scatter in the media, the differences in polarization feature imply differences in the microstructures. Polarization super-pixels obtained from Mueller images are clustered in polarization space, creating a map of tissue microstructural clusters. Our study reveals that the pixels from the sample clusters are spatially correlated, and each microstructural cluster corresponds with a specific pathological feature.

Several distinct clusters are observed, as shown in Fig. 3a. The clusters are clearly separated into two main groups, and detailed clusters can be seen in each group. Separated by a clear gap, two subgroups (clusters 1 and 2) are identified on the left, along with a few scattered clusters in the bottom left corner. The gap separating the clusters indicates that there is a distinct difference in polarization characteristic between cluster 1 and 2. On the other hand, the clusters on the right side are more convoluted compared to the left side. Cluster 6 is noticeably separated from the other clusters, but cluster 3, 4 and 5 are close to each other, with many visible subclusters. To interpret these clusters pathologically, we project the points from the polarization feature space onto the co-registered histological images. After the observation and evaluation by pathologists, the sample segmented structures from each polarimetric cluster are identified as displayed in Fig. 3a. Pathologists infer that the two clusters 1

and 2 on the left correspond to normal and cancerous cell nuclei, respectively. Furthermore, cluster 2 potentially correlates with differentiation degree, which will be discussed in the next section. Recall that the clear gap dividing cluster 1 and 2 indicates distinctive polarization characteristics, which in turn infer that the cancerous nuclei is microstructurally distinctive from benign nuclei. On the top-right corner, cluster 3 and 4 are involved with cell cytoplasm in liver H&E-stained pathological tissues. The two clusters are connected, indicating similarities in polarization feature, and the corresponding structural features, between the clusters. Unlike the other clusters, cluster 3 and 4 has many distinguishable detailed subsets. It suggests cell cytoplasm has various complicated subtypes of different microstructure or optical properties, which is plausible considering the rich variety in sizes and shapes for organelles within cells. Inferred by pathologists, cluster 5 and 6 mainly consist of collagen fiber and fibrocytes, respectively. In cluster 6, there are two identifiable subgroups, implying there are subtypes of fiber with different microstructure properties, based on their histological morphology. The unlabeled clusters correspond to noises or imaging artifacts. In short, the polarization pixels form clusters on the UMAP axes, and each cluster has a distinctive microstructure characterization. We discover that pixels clustered in polarization feature space are clustered spatially in histopathology images as well, segmenting pathologically meaningful structures such as normal and cancerous cell nucleus, cytoplasm, fibrocytes, and collagen fiber.

**Lesion-induced microstructural alteration**. Pathological alteration of tissues may induce changes in tissue microstructure, and here, we aim to visualize the microstructure transition at different stages of cellular differentiation. Specifically, cellular differentiation affects tissue microstructures, and such changes in microstructure composition can be revealed in the polarization feature space.

To explore the concept of visualizing microstructure transition during pathological changes, we collected HCC samples at different stages of differentiation degrees, and visualized the microstructure composition at each stage using a density heatmap on the UMAP atlas. Figure 3b demonstrates how ROIs from normal, well differentiation, moderate differentiation, and poor differentiation HCC samples differ in microstructural composition, represented by the UMAP density heatmap from the respective tissue samples. There seems to be a characteristic density distribution of polarization signatures at each stage of cellular differentiation degree, where the overall layout of the map remains stable, but the proportion and distribution of pixels that belongs to each cluster varies. The density ratios of several clusters appear to alter distinctly with different cellular differentiation degree: the density of cluster 2 (identified as cell nuclei by pathologists in pathological images) and cluster 6 (recognized as fibrocyte) both increase monotonically as the differentiation degree decreases. On the other hand, the lower right part of

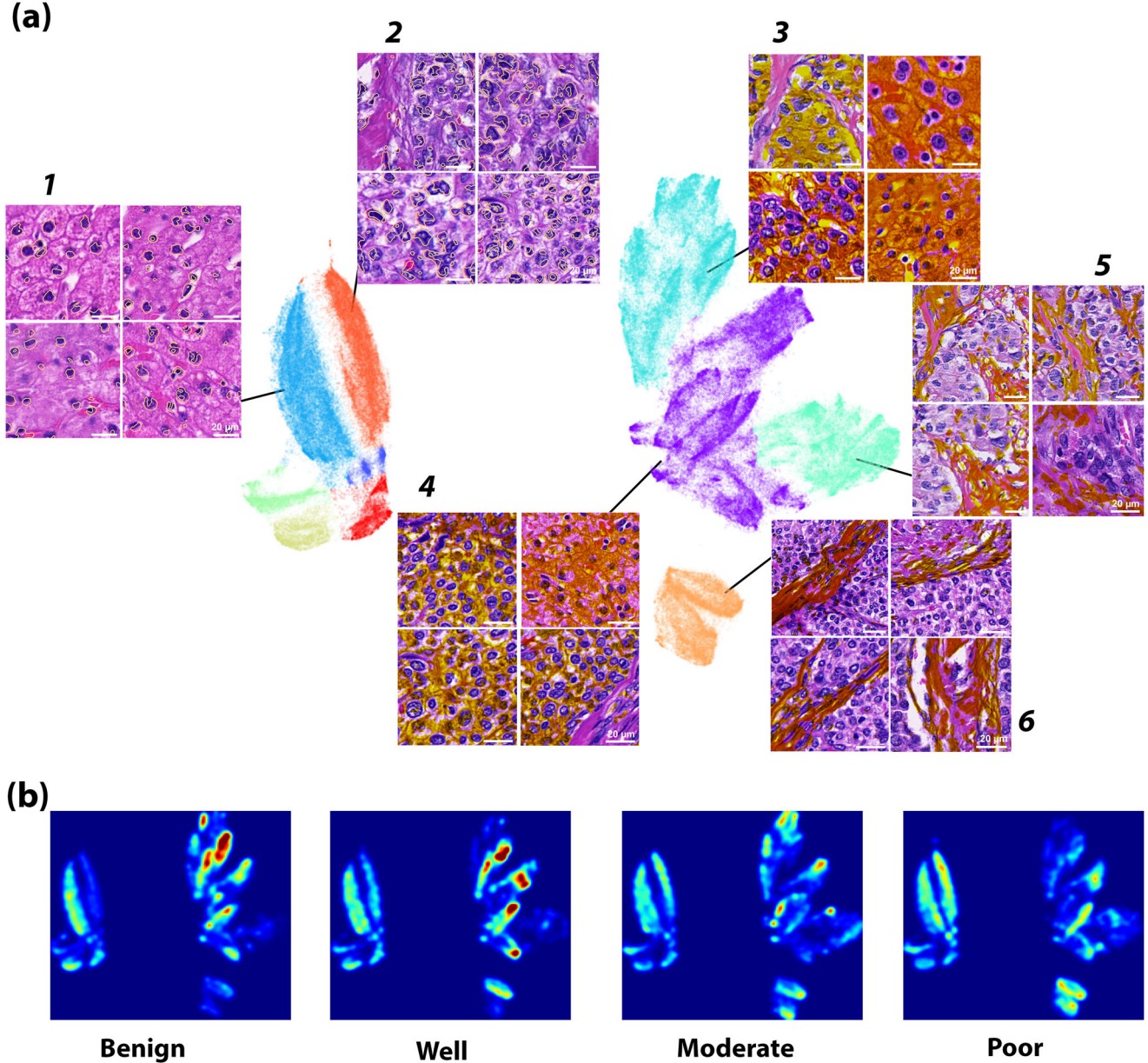

**Fig. 3 The UMAP clustering result of Mueller matrix pixel images and their corresponding pathological features.** The six polarization feature clusters segment pathological features on the histological images (**a**), and pathologists identify them as cell nucleus (cluster 1 and 2), cytoplasm (cluster 3 and 4), and collagen fiber and fibrocyte (cluster 5 and 6). The heatmap of the Uniform Manifold Approximation and Projection (UMAP) is drawn at various hepatocellular carcinoma (HCC) differentiation degrees (**b**), to help visualize variations in polarization features corresponding to pathological variation. An animated version of (**b**) is supplied in Supplementary Video 1.

cluster 3 (realized as cytoplasm) vanishes gradually as the differentiation degree decreases, while the upper left part of cluster 3 changes abruptly between well and moderately differentiated cellular states. Cluster 2 is clearly absent in benign tissues but present in malignant tissues, which implies that the presence of this specific microstructure characterization correlates strongly with tumor malignancy. Such polarization signature can be potentially used as the marker for HCC malignancy detection, providing a quantitative way to identify HCC tumor. The idea is thoroughly explored in the polarization marker section. Another point we noticed is that the configuration of the primary clusters is fixed, while the composition of clusters and the local density of data points varies with respect to cellular differentiation. Supplementary Video 1 is an animated version of Fig. 3b, showing the dynamical change in polarization composition as

normal cell develop to cancerous cell, and subsequently differentiate from well to poor differentiation degree, providing straightforward visualization of the difference in polarization composition at different stages of cellular differentiation. It potentially allows interpolation of polarization composition in between the states, such as between moderate and poor cellular differentiation degree. To shortly summarize: (1) polarization based microstructural clusters contain pathologically useful information such as cellular differentiation degree, and (2) the principal configuration of the microstructural map in UMAP representation remains stable; only the inter-cluster proportion changes with pathological variation.

**Cluster dendrogram**. Each cluster has its unique polarization characteristic and corresponding pathological feature, and the

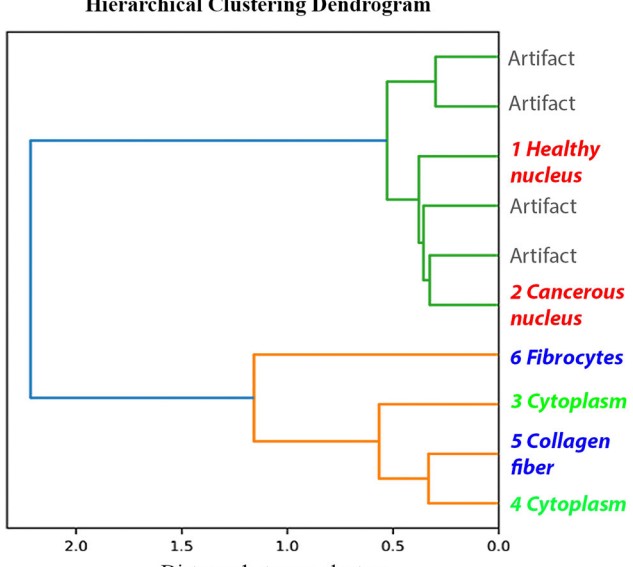

**Fig. 4 Dendrogram generated from hierarchical clustering process.** *X*-axis shows the distance between clusters, and the y-axis labels the different clusters and their corresponding pathological structures.

dendrogram plot in Fig. 4 illustrates how the pixels are separated into clusters using polarization features. Dendrogram illustrates how the clusters are formed, where the node indicates two clusters are merged into one, and the abscissa axis indicates the distance between clusters. Each branch corresponds to a certain variation in polarization features, and variation in polarization features infer variation in the microstructure. As shown in Fig. 4, the entire set first splits into two main groups, which contain nucleus structures (cluster 1, 2) and the rest (cluster 3, 4, 5, 6), including collagen fiber and cytoplasm. This indicates that cell nucleus and collagen fiber have very different microstructural properties, unsurprisingly. In the top strand, the two main nucleus clusters are extracted by stripping away the artifact clusters, mostly utilizing the equalities of Mueller matrices. In the bottom strand cluster 6 is the easiest to identify, as indicated by the between-cluster distance. The polarization microstructural map and dendrogram plot demonstrate that each subdivided cluster corresponds to a meaningful pathological microstructure in the H&E images, identifiable with distinctive polarization features.

**Polarization markers for diagnosis**. Tissue microstructure variation is pathologically meaningful. Here, we aim to extract polarization-based microstructure markers and explore their diagnostic value. Subwavelength microstructures, such as that of the cytoplasm, are potential polarization markers as well. Once the tissue microstructural subtypes are mapped using polarization features, it enables the visualization of pathological transition during primary liver cancer progression and the identification of potential polarization markers for tumor classification. To demonstrate, we attempt to tackle two different tasks: identification of HCC with different differentiation degrees, and classification of ICC from HCC.

We first study the microstructural variation during HCC progression and recognize cancerous tissues with different differentiation degrees. We observe the tissue microstructural composition alteration correlates with pathological variation in HCC, from normal liver tissues to well, moderate, and poor differentiation degree of HCC, which is clearly visualized in

Fig. 5a, and the animated heatmap provided in Supplementary Video 1. As indicated in Fig. 4 and Fig. 5a, we identified cluster 2 and 6 as the polarization marker to distinguish HCC from normal tissue. The cluster area proportion (area of pixels belonging to that specific cluster divided by the total area in the ROI) of cluster 2 and cluster 6 are calculated for normal tissue, highly differentiated, moderately differentiated, and poorly differentiated cancer samples, respectively. The resulting box-whisker plots are shown in Fig. 5b, and their corresponding p-values for the t-test are calculated to test for statistical significance. When using cluster 2 as the marker, the distinctions between neighboring sets are statistically significant, but not between the well differentiated and moderately differentiated samples ($p = 2.22 \times 10^{-5}$ for normal vs well, $p = 1.00 \times 10^{-4}$ for moderate vs poor, $p = 1.16 \times 10^{-1}$ for well vs moderate). The monotonically increasing trend of cluster 2 proportion is observed as well. To discriminate malignant HCC samples from normal tissues, we have separated all the samples into two sets, namely the malignant set (containing HCC samples of all three differentiation degrees) and the normal set (containing all the normal ROIs). Under this binary classification scenario, cluster 2 area proportion achieves an AUC of 94.84%. Using cluster 6 as the polarization marker shows a similar trend, as seen in Fig. 5c. The area proportions of cluster 6 in normal tissues are relatively low, most of which are under 5%. The separation between normal and highly differentiated samples ($p = 2.69 \times 10^{-2}$), and between moderately and highly differentiated samples ($p = 1.48 \times 10^{-4}$) are statistically significant, but not between highly and moderately differentiated samples ($p = 5.87 \times 10^{-1}$).

As labeled in Fig. 5d, we identified a cytoplasm structure in cluster 3 that is present in moderately differentiated samples, but not in highly differentiated samples. Using the labeled region as polarization marker, an AUC of 88.59% is achieved for distinguishing well differentiated from moderately differentiated samples, as shown in Fig. 5e. This observation is quite unconventional and requires validation in future works.

We now study the differences in tissue microstructural composition between HCC and ICC pathological samples, in an attempt to distinguish them. Likewise, we can first visualize the variation of tissue microstructural composition by observing the animated heatmap (provided in Supplementary Video 2) that samples gradually from HCC to ICC tissues, which provides clear visualization of the subtle difference in local density. The microstructures characterized by cluster 5, i.e., collagen fiber, is abundant in ICC samples, but not in HCC samples, as seen in Fig. 6a. In contrast, cluster 3, a subtype of cytoplasm, is abundant in HCC sample while not in ICC samples. Cluster 5 is the main focus as the polarization marker for HCC and ICC distinction. For comparison, Fig. 6b shows the box-whisker plot of the cluster 5 area proportion for both HCC and ICC samples. It is observed that most of HCC samples' cluster 5 proportion ratio is less than 10%, while that of the majority of ICC samples are above 10%. For discriminating ICC from HCC samples, the area proportion of cluster 5 yields an AUC of 84.94%. We also experimented with cluster 3, using it as the polarization marker yields an AUC of 71.69%, as seen in Fig. 6c. This implies that to a certain degree, the cytoplasm composition in HCC and ICC are different, and such difference can be reflected in the proportion of cluster 3, a polarization subtype of cytoplasm. It is noted that in this work identifications between the six polarization pixel clusters and their corresponding pathological composition were made by experienced pathologists based on color images of the H&E-stained slides, which show the boundaries of cells. Supplementary Figure 1 contains additional projection results onto high-definition H&E images, where the labeled region of cluster 3 is highlighted with brown color, similar to the color scheme of IHC.

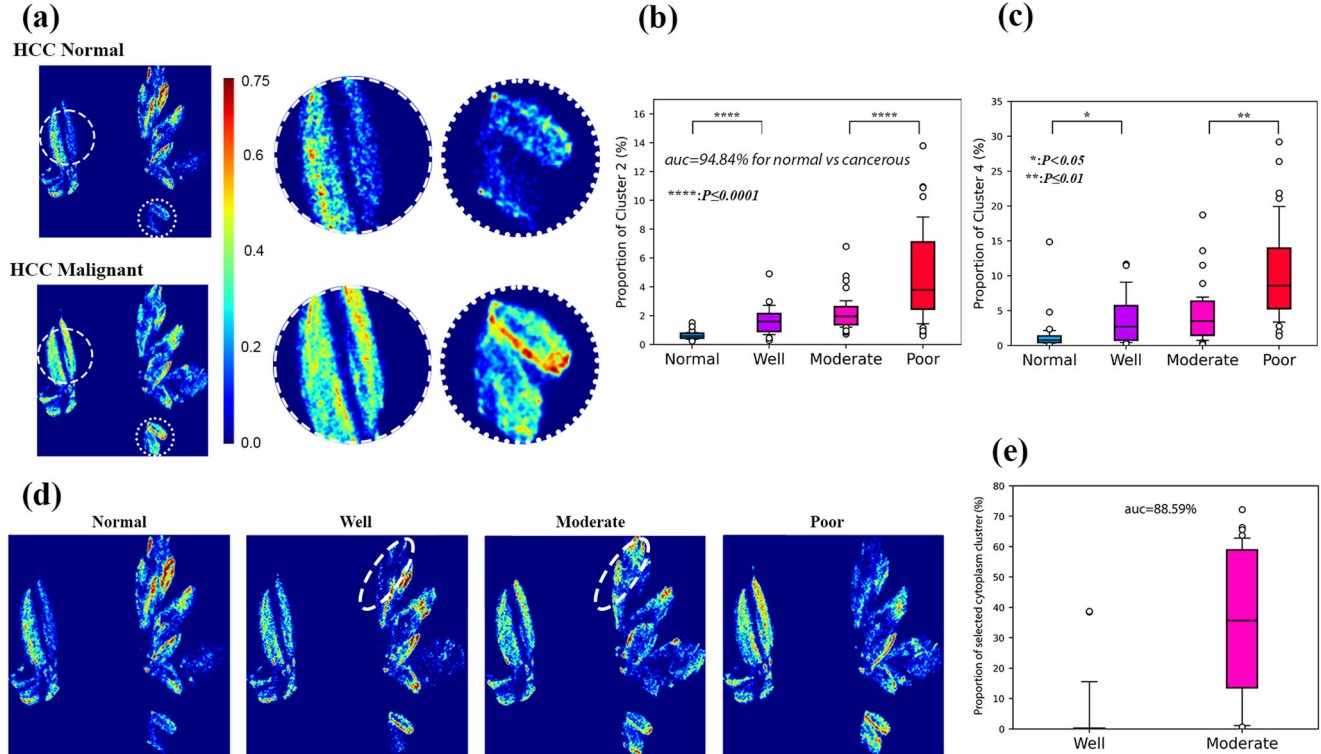

**Fig. 5 Identification of polarization markers that correlates with HCC differentiation degree.** The logarithm scale density heat maps of normal and malignant hepatocellular carcinoma (HCC) samples are compared in (**a**), and the two potential polarization markers are enlarged for better view. Proportion of pixels belongs to cluster 2 and 6 in each region of interest (ROI) are calculated, and the box whisker plots are shown in (**b**) and (**c**) for each differentiation degree. The box and the whiskers mark the 10, 25, 50, 75, and 90 percentiles respectively. **d** displays the heat map for each differentiation degree, and the dashed line indicates the selected polarization marker in the cytoplasm cluster to differentiate well and moderately differentiated samples. **e** This shows the box-whisker plot of the selected polarization marker in (**d**).

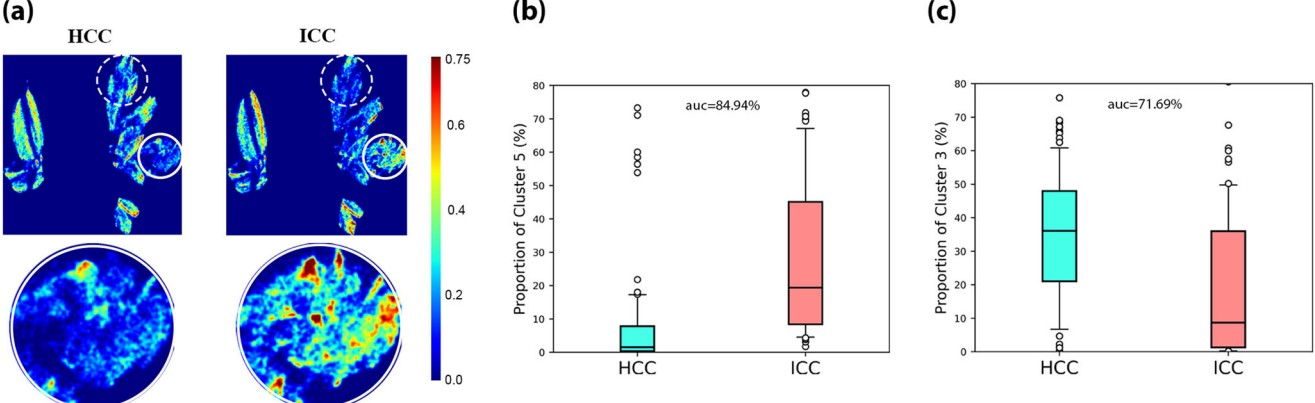

**Fig. 6 Identification of polarization marker for distinguishing HCC and ICC. a** It shows the logarithm scale density heat map for hepatocellular carcinoma (HCC) and intrahepatic cholangiocarcinoma (ICC) samples, and two clusters are labeled, one with solid line (cluster 5) and one with dashed line (cluster 3). The box whisker plots for the two polarization markers are shown in (**b**) and (**c**), the box and the whiskers mark the 10, 25, 50, 75, and 90 percentiles, respectively.

Supplementary Fig. 2 shows the projection results under even higher magnification (40×). Such identification between polarization and microstructural features will be further improved using other molecular specific staining methods or super-resolution techniques. While the current results are based on a relatively limited samples size, we aim to establish a correspondence between polarization features and microstructural variations in cytoplasm in future works.

## Discussion

Here, we have introduced a tissue microstructural composition analysis framework using Mueller microscopy and unsupervised learning methods. It is demonstrated that the resulting pixel clusters of Mueller images correspond to compositions of characteristic microstructure which contain pathologically useful information. We analyzed the cancerous tissues and the normal tissues around lesions in liver pathological samples, and split the

tissue structures into six components, each with its own distinctive polarimetric and microstructural feature. The clustered pixels in polarization feature space are correlated spatially, and by projecting the cluster labels onto histological images, we find that they segment various pathological structures. Based on the morphology on corresponding H&E images, pathologists identify that the six clusters can be respectively associated with the normal nucleus, cancerous nucleus, HCC and ICC cytoplasm, collagen fiber, and fibrocytes. While in this study the results are mainly compared against H&E staining, one can also conduct other types of staining that are specific to the tissue structure of interest for analysis. The difference in polarization characteristics implies a difference in microstructural characteristics. It is inferred that there are certain underlining microstructural differences between the identified clusters, causing their differences in polarization signature.

It is known that clustering results are sensitive to many different factors, including the experimented data, the used algorithm, and the parameters and hyperparameters. Figure 2 of this work also shows that a substantial data volume is needed to obtain a stable clustering. However, we are optimistic that the observed microstructural decomposition phenomenon using unsupervised learning method can be generalizable with further improvements in the algorithms and more priory information on the polarization and microstructural features of the samples. We find similar phenomenon in many other types of normal and cancerous tissues, such as the breast cancer and cervical intraepithelial neoplasia specimen shown in Supplementary Figs. 3 and 4. Due to the small sample size, we used the basic KMeans algorithm to cluster the polarization pixels into six clusters, and find a correspondence between the clusters and pathological structures, such as cell nuclei, cytoplasm, and collagen fiber.

We demonstrate how polarization markers can be extracted and potentially utilized for pathological applications. Tissue microstructural composition analysis enables visualization of tissue structure transition in polarization space during pathological variation. It is shown that cluster 2 correlates strongly with tumor. Evidence shows that cluster 2 can serve as a polarization marker to quantitatively characterize HCC samples with different differentiation degrees and separate HCC from normal liver tissues with an AUC of 95%. Differentiation of ICC from HCC is a known challenging pathological task. In this study, we identify a polarization marker to distinguish ICC from HCC on the H&E-stained slides, achieving an AUC of 84.94%. Through the transformation from polarization feature space to pathological feature space, it informs pathologists that cytoplasm and collagen contribute the most toward classification, which is consistent with the findings of medicine[47]. In particular, we observe that cluster 3 is a

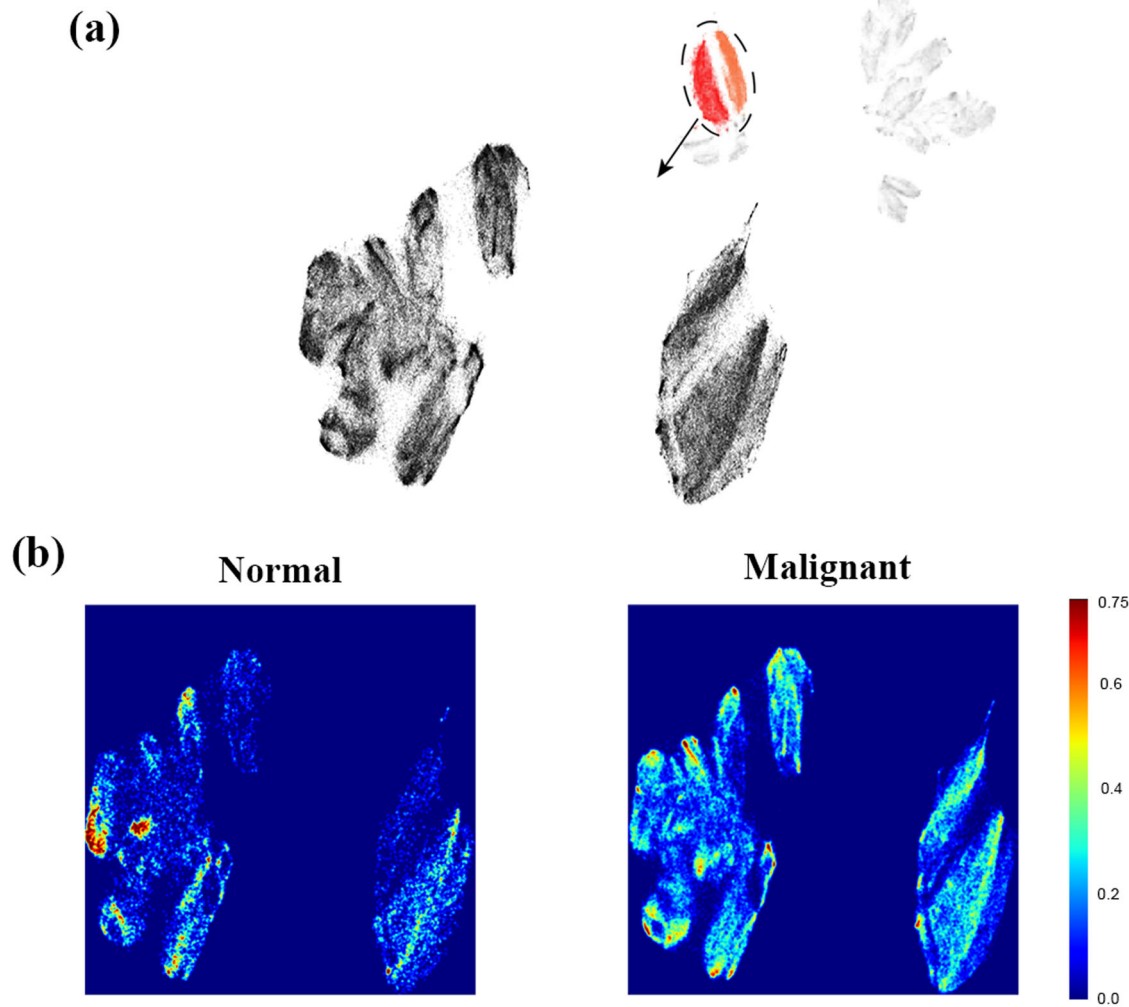

**Fig. 7 Clustering result of the cell nuclei clusters with finer super-pixels.** The pixels nucleus clusters, cluster 1 and 2, are collected and remapped using smaller sized super-pixels, as seen in (**a**). Detailed nucleus structures are revealed, two of which correlates with lesion malignant, as shown by the logarithm scale density plots in (**b**) of normal and malignant hepatocellular carcinoma (HCC) tissues.

polarization subtype of cytoplasm that is potentially meaningful for differentiating HCC and ICC, and the correspondence between polarization features and microstructural variations in cytoplasm may be further studied using molecular staining or super-resolution imaging methods. Therefore, this method may supplement the immunostaining analysis, providing interpretable and quantitative polarization markers for pathological diagnosis and prognosis prediction, and even reveal "invisible" super-resolution features to pathologists in an intuitive and visualized manner.

By adjusting the size of super-pixels and selecting interested microstructural clusters, it is possible to split an identified structure into even finer clusters, revealing detailed subcellular microstructures. To demonstrate, pixels from cluster 1 and 2 are collected and analyzed using the proposed method, only at a much smaller super-pixel size. As shown in Fig. 7a, the two clusters in the original UMAP are now transformed into four or more clusters. Figure 7b compares the heatmap between benign and malignant nuclei on the newly generated UMAP projection, it is clear that two distinct clusters correlate strongly with malignancy. Such a detailed pattern is only revealed after focusing on the nucleus structure and increasing the spatial resolution of polarimetric images.

In conclusion, by clustering the pixels of Mueller matrix images, we develop a tissue microstructural composition analysis framework to separate the intricate tissue structures into detailed microstructural components with distinctive polarization characteristics. Evidences show that the microstructure clusters are pathologically meaningful and their overall layout remain stable in polarization space with respect to samples from different individuals. The framework is applied to analyze the two most common primary liver cancers, HCC and ICC. The detailed map of biological microstructure in polarization space allowed the visualization of microstructure composition variation during HCC occurrence and progression, and the difference in microstructure between HCC and ICC. We also derived sets of polarization markers to tackle them quantitatively, yielding comparable performance with other state-of-the-art methods. Most importantly, even the "invisible" microstructure in pathology, the detailed structures of cytoplasm, could be presented in a "visible" manner and used for pathological diagnosis. Our results highlight the potential of polarization-based tissue compositional analysis framework as a tool to assist routine pathological tasks and explore the unknown pathological mechanism for pathologists and researchers.

## Data availability

The experimental data that support the findings of this study are available in Figshare with https://doi.org/10.6084/m9.figshare.23723490.

## Code availability

The code that supports the findings of this study is available from the corresponding author upon reasonable request.

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

## Acknowledgements

We thank Professor Jie Chen from Peking University, Chao He from Oxford University, Dr. Anli Hou and Dr. Shan Du from University of Chinese Academy of Sciences Shenzhen Hospital, for their invaluable feedback on the manuscript. H.M., J.W., Y.D., Y.Y., H.P., H.H., R.L. and N.Z. disclose support for the research of this work from National Natural Science Foundation of China [11974206, 61527826] and Shenzhen Bureau of Science and Innovation [JCYJ20170412170814624]. C.L., W.X., and R.P. disclose support for the publication of this work from Joint Funds for the Innovation of Science and Technology, Fujian province [2021Y9212]. R. H. disclose support for the publication of this work from Cross-disciplinary Research and Innovation Fund of SIGS [JC2022007].

## Author contributions

Conceptualization: J.W., Y.D., H.M., C.L. Methodology: J.W., Y.D., Y.Y., H.M., H.H., R.L., N.Z., H.P. Investigation: J.W., Y.D., H.M., C.L., R.H. Visualization: J.W., H.P. Supervision: H.M., C.L. Writing—original draft: J.W., Y.D., H.M., C.L. Writing—review & editing: H.M., C.L., R.H., J.X., Y.Y., W.X., R.P., X.T.

## Competing interests

The authors declare no competing interests.
