## [Peer Review File · Communications Engineering]

Reviewers' comments:

Reviewer #1 (Remarks to the Author):

The article by Wan et al. introduces a novel methodology to characterize pathological tissue based on Mueller Matrix microscopy.

The paper is well structured but could benefit from a more detail-oriented description of the methodology.

Major comments

- Methods

1. Which microscope has been used for the data acquisition? Is it the division of aperture or the homemade Mueller matrix system? If the system was the division of aperture, how did the authors deal with the reduction in extinction coefficient of the polarized camera? Was the system a reduced Mueller Matrix or a complete one?
2. Once the Mueller matrix images are obtained, a Gaussian filter is applied to increase the signal-to-noise ratio. This seems counterintuitive in an approach that is aimed at looking at microstructures. Could the authors comment on the type of filter and how the results improved with and without the filter?
3. Samples were acquired from 41 patients, a total of 222 ROIs are used for the unsupervised model construction. What were the dimensions of the ROI?
4. Was the thickness of the samples investigated to find an optimal one? As depolarization may drastically change the Mueller Matrix imagery.
5. The author compares Mueller Matrix images to H&E-stained pathological slides of liver cancer. Pathologists label the slides in a supervised training. Is H&E the only staining conducted? The assessment of collagen is not always specific in H&E yet cluster 5 and 6 are labeled as collagen and fibrocyte.
6. The description of the polarization super-pixel and polarimetric basis parameter $\begin{bmatrix} I \\ SEP \end{bmatrix}$ is rather vague, can the author be more specific?
7. The two supplementary videos do not seem to provide any useful information, can the author explain in more detail their purpose?
8. Figure 6, was any validation conducted on the cytoplasm composition?

Reviewer #2 (Remarks to the Author):

The authors propose an alternative approach that employs unsupervised pixel clustering of Mueller matrix images to group polarization pixels into clusters and suggests these clusters correspond to distinct pathological structures. Using hepatocellular carcinoma (HCC) and intrahepatic cholangiocarcinoma (ICC) samples, the authors demonstrate the proposed framework can be used for histological image segmentation, microstructure visualization, and marker identification for pathology variation differentiation. The study showcases the potential of unsupervised clustering in histopathological applications.

Major concerns are:

- 1) The authors need to clarify the major advantages of adopting Mueller matrix images for application, compared to polarization microscopy.

2) The study mainly analyzes the pixel clustering results and then utilizes the results to illustrate six associated structures. On the experimented data, the results seem separated and robust. While clustering results are affected by lots of factors, including the experimented data, the used algorithm, and the parameters and hyperparameters. With limited experimented data from only one institute generated under similar settings, the generalization capability of clustering results to other sources of data needs further experimentation and validation.

3) Without open-source data and implementation, not sure about the reproducibility of the results.

Reviewer #1 (Remarks to the Author):

The article by Wan et al. introduces a novel methodology to characterize pathological tissue based on Mueller Matrix microscopy.

The paper is well structured but could benefit from a more detail-oriented description of the methodology.

Major comments

- Methods

1. Which microscope has been used for the data acquisition? Is it the division of aperture or the homemade Mueller matrix system? If the system was the division of aperture, how did the authors deal with the reduction in extinction coefficient of the polarized camera? Was the system a reduced Mueller Matrix or a complete one?

Reply:

Thank you for these questions. We used two types of Mueller microscopes, one of which used the division of focal plane polarimeters (DoFP) for 2D Stokes imaging. The extinction coefficient of the DoFP is 150 at 633nm wavelength. An optimization and calibration algorithm was used to minimize the average root mean square error. The following modification are made in the Method section:

“Histological slides obtained from biopsy are measured using Mueller matrix microscopes (MMM) under 20x 0.4NA objective lens *with 633nm wavelength to obtain the complete Mueller matrix (4x4) images*. The choice of this moderate NA allows us to achieve spatial resolution of 1 micron for resolving subcellular structures but maintain optimal accuracy in the Mueller matrix measurements. The homemade Mueller matrix microscopes *were* upgraded from low-cost commercial upright transmission microscopes by *adding polarization optics modules in the existing optical path*. *Most of the samples were imaged by a MMM with a pair of fixed polarizer and rotating waveplate for both the polarization state generator and analyzer^{10,23}, and a 1001x1301 pixels 12-bit CCD camera (74-0107A, QImaging, Canada)*. It is capable of taking a Mueller matrix image in about 3 minutes with less than 0.02 average root mean square error (RMSE) *using standard samples including air, linear polarizer, and waveplate*. Details for optimization and calibration of the microscope can be found in previous studies ^{10,23-25}. *A small number of samples were imaged by an upgraded MMM²⁶ whose polarization state analyser and 2D detector were replaced by two 2048 × 2448 pixels 16 bit division of focal plane (DoFP) polarimeters (PHX050S-PC, Lucid Vision Labs Inc., Canada). Extinction coefficient of the DoFP is 150 at 633nm wavelength. The new MMM is capable of more than a factor of 10 increase in speed and a factor of 2 improvement in average RMSE, compared with rotating-waveplate MMM. Detailed specifications of both MMMs can be found in reference 26*. RMSE can be reduced further by rotating both the polarizer and the wave plate in the polarization state generator ²⁷.”

Reference:

26 Huang, T. *et al.* Fast Mueller matrix microscope based on dual DoFP polarimeters. *Opt. Lett.* **46**, 1676-1679 (2021).

2. Once the Mueller matrix images are obtained, a Gaussian filter is applied to increase

the signal-to-noise ratio. This seems counterintuitive in an approach that is aimed at looking at microstructures. Could the authors comment on the type of filter and how the results improved with and without the filter?

Reply:

Thank you for your suggestion. In contrast to conventional imaging methods, Mueller imaging methods obtain much richer microstructural information at pixel level by extracting relevant polarization features. To help clarify, we made the following modification to the manuscript:

“ Once the Mueller matrix images are obtained, a Gaussian filter is applied to *reduce pixel value fluctuations while preserving the polarization features of microstructures, because polarization features is observed to be insensitive to image resolution*¹⁵. Through the sum decomposition of Mueller matrix introduced by Cloude, the physically unrealizable part of Mueller matrices is filtered ²⁸. ”

15. Yudi Liu, Yang Dong, Lu Si, Ruoyu Meng, Yanmin Dong, and Hui Ma*. “Comparison between image texture and polarization features in histopathology”, Biomedical Optics Express, 12(3): 1593-1608, 2021

3. Samples were acquired from 41 patients, a total of 222 ROIs are used for the unsupervised model construction. What was the dimensions of the ROI?

Reply:

Thank you for your question. The dimension for most of the ROIs is 1001x1301 pixels, but 2048 × 2448 for a small number of ROIs. We have modified the manuscript to include description regarding the dimensions of the ROI, as shown in the Reply to question 1.

4. Was the thickness of the samples investigated to find an optimal one? As depolarization may drastically change the Mueller Matrix imagery.

Reply:

Thank you for your question. We used standard pathological slide which is usually of 4-micron thickness. The manuscript is modified to address sample thickness in Methods section regarding samples:

“ Hepatocellular carcinoma (HCC) accounts for 70% of primary liver tumor, one of the cancers with the worst survival rate (20%) ¹⁸. Undoubtedly, accurate staging of HCC subtypes has significant clinical implications. As the second most common type of primary liver tumor, intrahepatic cholangiocarcinoma (ICC) is much more aggressive than HCC, and the differentiation between them remains challenging ¹⁹⁻²². Here, we use HCC and ICC samples to demonstrate the validity of our proposed method. Pathological slides of all patients are provided by Fujian Medical University Cancer Hospital. Sampled from 41 patients, a total of 222 ROIs are used for the unsupervised model construction, in which 181 ROIs are labeled. Among them, 71 ROIs are from ICC samples, 85 ROIs are from HCC samples, and 25 ROIs are from normal tissues around lesions. *The thickness of pathological slides are around 4 microns, which is within the single scattering regime and usually*

corresponds to small depolarization. The study was approved by the Ethics Committee of Fujian Medical University Cancer Hospital, and participants provided written informed consent to take part in the study. ”

5. The author compare Mueller Matrix images to H&E-stained pathological slides of liver cancer. Pathologist label the slides in a supervised training. Is H&E the only staining conducted? The assessment of collagen is not always specific in H&E yet cluster 5 and 6 are labeled as collagen and fibrocyte.

Reply:

Thank you for this question, we have modified the manuscript in the Results and Discussion section to address the point:

In the Results section:

“ ... On the top-right corner, cluster 3 and 4 are involved with cell cytoplasm in liver H&E-stained pathological tissues. The two clusters are connected, indicating similarities in polarization feature, and the corresponding structural features, between the clusters. Unlike the other clusters, cluster 3 and 4 has many distinguishable detailed subsets. It suggests cell cytoplasm has various complicated subtypes of different microstructure or optical properties, which is plausible considering the rich variety in sizes and shapes for organelles within cells. *Inferred by pathologists*, cluster 5 and 6 mainly consist of collagen fiber and fibrocytes respectively, *based on their histological morphology*. In cluster 6, there are two identifiable subgroups, implying there are subtypes of fiber with different microstructure property. The unlabeled clusters correspond to noises or imaging artifacts. In short, the polarization pixels form clusters on the UMAP axes, and each cluster has a distinctive microstructure characterization. We discover that pixels clustered in polarization feature space are clustered spatially in histopathology images as well, segmenting pathologically meaningful structures such as normal and cancerous cell nucleus, cytoplasm, fibrocytes, and collagen fiber.”

And in the Discussion section:

“Here, we have introduced a tissue microstructural composition analysis framework using Mueller microscopy and unsupervised learning methods. It is demonstrated that the resulting pixel clusters of Mueller images correspond to compositions of characteristic microstructure which contain pathologically significant information. We analyzed the cancerous tissues and the normal tissues around lesions in liver pathological samples, and split the tissue structures into 6 components, each with its own distinctive polarimetric and microstructural feature. The clustered pixels in polarization feature space are correlated spatially, and by projecting the cluster labels onto histological images we find that they segment various pathological structures. *In particular, Based on the morphology on corresponding H&E images, pathologists identify that* the six clusters can be respectively associated with the normal nucleus, cancerous nucleus, HCC and ICC cytoplasm, collagen fiber, and fibrocytes. *While in this study the results are mainly compared against H&E staining, one can also conduct other types of staining that are specific to the tissue structure of interest for analysis.* The difference in polarization characteristics implies a difference in

microstructural characteristics. It is inferred that there are certain underlining microstructural differences between the identified clusters, causing their differences in polarization signature.”

6. The description of the polarization super-pixel and polarimetric basis parameter is rather vague, can the author be more specific?

Reply:

Thank you for your constructive suggestion. As per suggested, a more detailed description of polarization super-pixel and the polarimetric basis parameters are added to Method section. The super-pixel section is reorganized and rewritten to clarify the implementation and motivation of polarization super-pixel.

In the Method section, the section name “Data preprocessing: polarization super-pixel” is changed to “Polarization super-pixel”. In the meantime, the first paragraph of the “Data preprocessing: polarization super-pixel” section is moved to “Mueller matrix images” section. The manuscript is modified as the following:

“ Mueller matrix images.

Histological slides obtained from biopsy are measured using Mueller matrix microscope under 20x 0.4NA objective lens. The choice of this moderate NA allows us to achieve spatial resolution of 1 micron for resolving subcellular structures but maintain optimal accuracy in the Mueller matrix measurements. The homemade Mueller matrix microscope was upgraded from a low-cost commercial upright transmission microscope by installing polarization state generator and analyzer in the existing optical path, based on fixed polarizer and rotating waveplate setup^{10,23}. It is capable of taking a Mueller matrix image in about 3 minutes with less than 0.02 average root mean square error (RMSE). Details for optimization and calibration of the microscope can be found in previous studies^{10,23-25}. Recently, the development of DoFP (division of focal plane) Mueller microscope²⁶ pushes both imaging speed and accuracy to the next level. Compared to the dual waveplate method, DoFP method’s acquisition time decreased by more than a factor of 10 while improving the RMSE by a factor of 2²⁶. RMSE can be reduced further by rotating both the polarizer and the wave plate in the polarization state generator²⁷. Taking advantage of such instrumental advancement, DoFP Mueller microscope is used in complement to the dual rotating waveplate Mueller microscope during polarimetric data acquisition.

Once the Mueller matrix images are obtained, a Gaussian filter is applied to reduce pixel value fluctuations while preserving the polarization features of microstructures, because polarization features is observed to be insensitive to changes in image resolution¹⁵. Through the sum decomposition of Mueller matrix introduced by Cloude, the physically unrealizable part of Mueller matrices is filtered²⁸.

Polarization super-pixel.

Super-pixel approach is a proven method to reduce data redundancy while decreasing computational complexity for downstream tasks in the field of image processing

²⁹⁻³². Considering the large number of polarization pixels to be processed, a polarization super-pixel strategy is taken in our scheme to compress the volume of polarimetric data while preserving its main polarization features¹³.

Specifically, the polarization super-pixels are created with the following steps in this work. First of all, we standardize the Mueller matrix elements of individual polarization pixels by removing the mean and scaling them to unit variance⁴⁷. Then we apply the minibatch KMeans algorithm³³ to all the polarization pixels in the ROI and group them into 1024 clusters, using the 15 Mueller matrix elements normalized by M11 as features. Lastly, the average of the polarization basis parameters¹⁷ are calculated and recorded for each polarization super-pixel. Note that we can trace back the coordinates of the pixels that belongs to the specific super-pixel.

Instead of preserving the entirety of local information by considering all the pixels, we characterize the local subset of neighboring points in polarization feature space with their centroids, and perform unsupervised algorithms on the set of centroids. Note that the spatial constraint of generic super-pixels is waived since it increases the compression rate for polarization data. Using the described super-pixel approach, pixels with similar polarization characterization are grouped, creating N polarization super-pixels for each ROI. N can be chosen based on the complexity of the images in polarization space. A larger N preserves more polarization information, while a smaller N leads to more compact data representation. Effectively, computing at the scale of super-pixel level instead of pixel level allows us to ignore the detailed local structure of polarization data and focus on the global structure in polarization space, improving robustness of the result. Through the aggressive use of super-pixels, the data volume is decreased by 3 orders of magnitude.

47. Grus, J. (2015). *Data science from scratch: first principles with Python. First edition.* Sebastopol, CA, O'Reilly.

Data preprocessing: polarization super-pixel.

Once the Mueller matrix images are obtained, a Gaussian filter is applied to increase the signal-to-noise ratio. Through the sum decomposition of Mueller matrix introduced by Cloude, the physically unrealizable part of Mueller matrices is filtered²⁸.

Considering the large number of polarization pixels to be processed, a multi-scale strategy is taken in our scheme. Instead of preserving the entirety of local information by considering all the pixels, we characterize the local subset of neighboring points in polarization feature space with their centroids, and perform unsupervised algorithms on the set of centroids. Effectively, computing at the scale of super-pixel level instead of pixel level allows us to ignore the detailed local structure of polarization data and focus on the global structure in polarization space, improving robustness of the result. Such super-pixel approach is a proven method to reduce data redundancy while decreasing computational complexity for downstream tasks²⁹⁻³². On demand, the super-pixel can be unpacked and analyzed for more detailed features once the super-pixels of interests are selected.

Using the described super-pixel approach, pixels with similar polarization characterization are grouped, creating N polarization super-pixels for each ROI. Note that the spatial constraint of generic super-pixels is waived since it increases the compression rate for polarization data. N can be chosen based on the complexity of the images in polarization space. A larger N preserves more polarization information, while a smaller N leads to more compact data representation. Since Mueller matrix is a comprehensive

representation of polarimetric data, the super-pixels are created based on the Mueller matrix images combined with the minibatch KMeans algorithm³³. The super-pixel method is utilized to compress the volume of polarimetric data while preserving its main polarization features¹³. It is otherwise impossible to process the vast amount of pixel-level data points with unsupervised techniques. Through the aggressive use of super-pixels, the data volume is decreased by 3 orders of magnitude. ”

And in the Method section, polarimetric basis parameters:

“Mueller matrix describes comprehensively the sample’s polarization properties, which provides extensive microstructural information. However, the explicit relationships between Mueller matrix elements and the characteristic microstructures are usually unclear, and many of the elements are sensitive to sample orientation. To accommodate this shortcoming, various decomposition³⁴⁻³⁶ and transformation³⁷⁻⁴⁰ methods have been proposed to provide polarization parameters which are functions of Mueller matrix elements but tend to have more explicit interpretability from the perspective of physics. These polarization parameters are used as the polarization basis parameters (PBP) for further statistical analysis^{11,14, 17}. *A comprehensive list of the used PBPs are provided in supplementary table 1. More details are also available from reference^{17,37}”*

The full list of used polarimetric basis parameter is added to the supplementary document:

Supplementary table 1. Full list and description of the polarimetric basis parameters.

Polarimetric basis parameter	Description or explicit formula
B_b	$(M_{22} + M_{33})/2$
B_β	$(M_{23} - M_{32})/2$
$B_{\bar{b}}$	$(M_{22} - M_{33})/2$
$B_{\bar{\beta}}$	$(M_{23} + M_{32})/2$
t_1	$\sqrt{B_b^2 + B_{\bar{\beta}}^2}/2$
$\det B$	$M_{22}M_{33} - M_{23}M_{32}$
$\text{norm } B$	$\sqrt{(M_{22}^2 + M_{33}^2 + M_{23}^2 + M_{32}^2)}$
CD	$M_{14} + M_{41}$
P_L	$\sqrt{M_{21}^2 + M_{31}^2}$
D_L	$\sqrt{M_{12}^2 + M_{13}^2}$
r_L	$\sqrt{M_{24}^2 + M_{34}^2}$
q_L	$\sqrt{M_{42}^2 + M_{43}^2}$

Transpose asym DP	$\sqrt{(M_{12} - M_{21})^2 + (M_{13} - M_{31})^2}$
Transpose asym rq	$\sqrt{(M_{24} + M_{42})^2 + (M_{34} - M_{43})^2}$
det MM	Matrix determinant of Mueller matrix
norm MM	Matrix norm of Mueller matrix
Trace MM	Trace of Mueller matrix
$ \vec{P} $	$\sqrt{M_{21}^2 + M_{31}^2 + M_{41}^2}$
$ \vec{D} $	$\sqrt{M_{12}^2 + M_{13}^2 + M_{14}^2}$
$\vec{P} \cdot \vec{D}$	$M_{12}M_{21} + M_{13}M_{31} + M_{14}M_{41}$
$\vec{P}m\vec{D}$	Invariant under rotation and retarder transformation
$\vec{P}m^T\vec{D}$	Invariant under rotation and retarder transformation
λ_1	Eigenvalues of coherence matrix
λ_2	Eigenvalues of coherence matrix
λ_3	Eigenvalues of coherence matrix
λ_4	Eigenvalues of coherence matrix
P_1	Indices of polarimetric purity
P_2	Indices of polarimetric purity
P_3	Indices of polarimetric purity
PI	Overall purity index
PD	Depolarization index
S	Polarization entropy
ES	$M_{11} - M_{22} - M_{33} + M_{44}$
E_1	$(M_{11} + M_{22})^2 - (M_{12} + M_{22})^2 - (M_{33} + M_{44})^2 - (M_{34} - M_{43})^2$
E_2	$(M_{11} - M_{22})^2 - (M_{12} - M_{21})^2 - (M_{33} - M_{44})^2 - (M_{34} + M_{43})^2$
E_3	$(M_{11} + M_{21})^2 - (M_{12} + M_{22})^2 - (M_{13} + M_{23})^2 - (M_{14} + M_{24})^2$
E_4	$(M_{11} - M_{21})^2 - (M_{12} - M_{22})^2 - (M_{13} - M_{23})^2 - (M_{14} - M_{24})^2$
E_5	$(M_{11} + M_{12})^2 - (M_{21} + M_{22})^2 - (M_{31} + M_{32})^2 - (M_{41} + M_{42})^2$

E_6	$(M_{11} - M_{12})^2 - (M_{21} - M_{22})^2$ $- (M_{31} - M_{32})^2$ $- (M_{41} - M_{42})^2$
D	Diattenuation
Δ	Average of the principle depolarization
δ	Retardation angle of linear retardance
R	Total retardance
α	Angle of optical rotation

7. The two supplementary videos do not seem to provide any useful information can the author explain in more detail their purpose?

Reply:

Thank you for this suggestion. To emphasize the information provided in supplementary videos, the following modification to the manuscript is made in the Results section:

“ To explore the concept of visualizing microstructure transition during pathological changes, we collected HCC samples at different stages of differentiation degrees, and visualized the microstructure composition at each stage using a density heatmap on the UMAP atlas. Fig.3b demonstrates how ROIs from normal, well differentiation, moderate differentiation, and poor differentiation HCC samples differ in microstructural composition, represented by the UMAP density heatmap from the respective tissue samples. There seems to be a characteristic density distribution of polarization signatures at each stage of cellular differentiation degree, where the overall layout of the map remains stable, but the proportion and distribution of pixels that belongs to each cluster varies. The density ratios of several clusters appear to alter distinctly with different cellular differentiation degree: the density of cluster 2 (identified as cell nuclei by pathologists in pathological images) and cluster 6 (recognized as fibrocyte) both increase monotonically as the differentiation degree decreases. On the other hand, the lower right part of cluster 3 (realized as cytoplasm) vanishes gradually as differentiation degree decreases, while the upper left part of cluster 3 changes abruptly between well and moderately differentiated cellular states. Cluster 2 is clearly absent in benign tissues but present in malignant tissues, which implies that the presence of this specific microstructure characterization correlates strongly with tumor malignancy. Such polarization signature can be potentially used as the marker for HCC malignancy detection, providing a quantitative way to identify HCC tumor. The idea is thoroughly explored in the polarization marker section. Another point we noticed is that the configuration of the primary clusters is fixed, while the composition of clusters and the local density of data points varies with respect to cellular differentiation. *Supplementary video 1 is an animated version of Fig. 3b, showing the dynamical change in polarization composition as heathy cell develop to cancerous cell, and subsequently differentiate from well to poor differentiation degree, providing straight forward visualization of the difference in polarization composition at different stages of cellular differentiation. It potentially allows*

interpolation of polarization composition in between the states, such as between moderate and poor cellular differentiation degree. To shortly summarize: (1) polarization based microstructural clusters contain pathologically significant information such as cellular differentiation degree, and (2) the principal configuration of the microstructural map in UMAP representation remains stable; only the inter-cluster proportion changes with pathological variation. ”

“ We now study the differences in tissue microstructural composition between HCC and ICC pathological samples, in an attempt to distinguish them. Likewise, we can first visualize the variation of tissue microstructural composition by observing the animated heatmap (provided in *Supplementary video 2*) that samples gradually from HCC to ICC tissues, *which provides clear visualization of the subtle difference in local density.* The microstructures characterized by cluster 5, i.e., collagen fiber, is abundant in ICC samples, but not in HCC samples, as seen in Fig.6a. In contrast, cluster 3, a subtype of cytoplasm, is abundant in HCC sample while not in ICC samples. Cluster 5 is the main focus as the polarization marker for HCC and ICC distinction. For comparison, Fig.6b shows the box-whisker plot of the cluster 5 area proportion for both HCC and ICC samples. It is observed that most of HCC samples’ cluster 5 proportion ratio is less than 10%, while that of the majority of ICC samples are above 10%. For discriminating ICC from HCC samples, the area proportion of cluster 5 yields an AUC of 84.94%. We also experimented with cluster 3, using it as the polarization marker yields an AUC of 71.69%, as seen in Fig.6c. This implies that to a certain degree, the cytoplasm composition in HCC and ICC are different. ”

8. Figure 6, was any validation conducted on the cytoplasm composition?

Reply:

Thank you for your question. In this work, cytoplasm composition was identified by pathologists from the color images of the H&E-stained slides, which show the boundaries of cells. To clarify, we have edited the manuscript as the following in the Results section:

“We now study the differences in tissue microstructural composition between HCC and ICC pathological samples, in an attempt to distinguish them. Likewise, we can first visualize the variation of tissue microstructural composition by observing the animated heatmap (provided in *Supplementary 2*) that samples gradually from HCC to ICC tissues, which provides clear visualization of the subtle difference in local density. The microstructures characterized by cluster 5, i.e., collagen fiber, is abundant in ICC samples, but not in HCC samples, as seen in Fig.6a. In contrast, cluster 3, a subtype of cytoplasm, is abundant in HCC sample while not in ICC samples. Cluster 5 is the main focus as the polarization marker for HCC and ICC distinction. For comparison, Fig.6b shows the box-whisker plot of the cluster 5 area proportion for both HCC and ICC samples. It is observed that most of HCC samples’ cluster 5 proportion ratio is less than 10%, while that of the majority of ICC samples are above 10%. For discriminating ICC from HCC samples, the area proportion of cluster 5 yields an AUC of 84.94%. We also experimented with cluster 3, using it as the polarization marker yields an AUC of 71.69%, as seen in Fig.6c. This implies that to a

certain degree, the cytoplasm composition in HCC and ICC are different. *It is noted that in this work identifications between the six polarization pixel clusters and their corresponding pathological composition were made by pathologists based on color images of the H&E stained slides, which show the boundaries of cells. Such identification between polarization and microstructural features will be further improved using other molecular specific staining methods or super-resolution techniques. ”*

Reviewer #2 (Remarks to the Author):

The authors propose an alternative approach that employs unsupervised pixel clustering of Mueller matrix images to group polarization pixels into clusters and suggests these clusters correspond to distinct pathological structures. Using hepatocellular carcinoma (HCC) and intrahepatic cholangiocarcinoma (ICC) samples, the authors demonstrate the proposed framework can be used for histological image segmentation, microstructure visualization, and marker identification for pathology variation differentiation. The study showcases the potential of unsupervised clustering in histopathological applications.

Major concerns are:

1. The authors need to clarify the major advantages of adopting Mueller matrix images for application, compared to polarization microscopy.

Reply:

Thank you for your comment, we agree that it is important to layout the main advantage of using Mueller matrix microscope over polarization microscope, especially in our specific application scenario. The introduction section of the manuscript is edited as follows:

“ It has been known that interactions between polarized photons and complex media encode comprehensive microstructural information. *The 4x4 polarization transformation matrix, i.e. Mueller matrix, provide a comprehensive representation on the specimen's polarimetric properties, while non-polarization or other polarization methods only reveal a subset of them. Since* polarization properties are closely related to the microstructural features *of scattering media*^{6,7}, particularly *sensitive to* super resolution features at subwavelength scale^{8,9}, Mueller matrix microscopy is capable of mapping tissue architecture down to subcellular level^{9,10}. Being treated as an inverse problem, the pixel-level correlation between microstructure and polarization features are usually established by taking supervised learning approaches¹¹⁻¹⁴, as illustrated in Fig.1. Pathologist segments regions on the H&E-stained pathological images to provide spatial labeling of the interested pathological structures, which in turn induces labeled pixels on the co-registered polarization images as the ground truth. Then polarization features corresponding to the labeled pixels are identified using supervised machine learning techniques. Such analysis can be performed on a low-magnification optical system, based on the finding that polarimetric imaging's contrast mechanism is insensitive to image resolution¹⁵. This approach has been used to study a variety of biological tissue samples through the extraction of microstructure-specific polarization features^{11,14,16,17}. In particular, microstructures in liver tissue during lesion have been analyzed to extract the simplest form of polarization feature parameter for cancerous cell nucleus identification in hepatocellular carcinoma samples¹⁶. ”

2. The study mainly analyzes the pixel clustering results and then utilizes the results to illustrate six associated structures. On the experimented data, the results seem separated and robust. While clustering results are affected by lots of factors, including the experimented data, the used algorithm, and the parameters and hyperparameters. With

limited experimented data from only one institute generated under similar settings, the generalization capability of clustering results to other sources of data needs further experimentation and validation.

Reply:

Thank you for your thoughtful comment. We agree that the manuscript lacks discussion on the topic of generalizability capability, so we have edited the discussion section of the manuscript as following:

“ Here, we have introduced a tissue microstructural composition analysis framework using Mueller microscopy and unsupervised learning methods. It is demonstrated that the resulting pixel clusters of Mueller images correspond to compositions of characteristic microstructure which contain pathologically significant information. We analyzed the cancerous tissues and the normal tissues around lesions in liver pathological samples, and split the tissue structures into 6 components, each with its own distinctive polarimetric and microstructural feature. The clustered pixels in polarization feature space are correlated spatially, and by projecting the cluster labels onto histological images we find that they segment various pathological structures. Based on the morphology on corresponding H&E images, pathologists *identified* that the six clusters can be respectively associated with the normal nucleus, cancerous nucleus, HCC and ICC cytoplasm, collagen fiber, and fibrocytes. While in this study the results are mainly compared against H&E staining, one can certainly conduct other types of staining that are specific to the tissue structure of interest for analysis. The difference in polarization characteristics implies a difference in microstructural characteristics. It is inferred that there are certain underlining microstructural differences between the identified clusters, causing their differences in polarization signature.

It is known that clustering results are sensitive to many different factors, including the experimented data, the used algorithm, and the parameters and hyperparameters. Figure 2 of this work also show that a substantial data volume is needed to obtain a stable clustering. However, we are optimistic that the observed microstructural decomposition phenomenon using unsupervised learning method can be generalizable with further improvements in the algorithms and more priory information on the polarization and microstructural features of the samples. We find similar phenomenon in many other types of normal and cancerous tissues, such as the breast cancer and cervical cancer specimen shown in supplementary figure 1 and 2. Due to the small sample size, we used the basic KMeans algorithm to cluster the polarization pixels into 6 clusters, and find a correspondence between the clusters and pathological structures, such as cell nuclei, cytoplasm, and collagen fiber.”

Supplementary figure 1:

Supplementary figure 1. KMeans clustering result on breast cancer specimen, with the KMeans cluster labels, corresponding H&E images, and two zoomed in regions for comparison.

Supplementary figure 2:

Supplementary figure 2. KMeans clustering result on cervical cancer specimen, with the KMeans cluster labels, corresponding H&E images, and two zoomed in regions for comparison.

3. Without open-source data and implementation, not sure about the reproducibility of the results.

Reply:

Thank you for your comment. To help reproduce our results, we are willing to open-source our data, including the Mueller matrix data, available corresponding H&E images and M11 intensity image, and the meta-label of cancer/tissue type. We believe that open sourcing the dataset will increase the reproducibility of our results and make our work accessible to the research community, who can then build upon our findings. In our manuscript, one can find the detailed description of our implementation as well.

The data availability statement is modified as following:

Data availability ~~The data that support the findings of this study are available from the corresponding author upon reasonable request.~~ The experimental data that support the findings of this study are available in Figshare with DOI: [10.6084/m9.figshare.23723490](https://doi.org/10.6084/m9.figshare.23723490). We encourage interested researchers to discuss and share the data analysis results with us.

Reviewers' comments:

Reviewer #1 (Remarks to the Author):

The authors provided extensive replies to my original questions but most of the responses are still speculative.

For example in question # 5 I asked what type of validation the author had for collagen (meaning SHG or other staining) the reply was, our pathologists marked it as collagen.

Similarly, for the cytoplasm composition question # 8, the authors have no validation.

Considering the fact that small variations in tissue composition of ANY KIND can alter the Mueller Matrix results I believe that a validation experiment is needed.

Reviewer #2 (Remarks to the Author):

The rebuttal addressed most of my concerns, and the current manuscript is clearer and more scientifically rigorous.

Reviewer #1 (Remarks to the Author):

The authors provided extensive replies to my original questions but most of the responses are still speculative.

For example in question # 5 I asked what type of validation the author had for collagen (meaning SHG or other staining) the reply was, our pathologists marked it as collagen. Similarly, for the cytoplasm composition question # 8, the authors have no validation. Considering the fact that small variations in tissue composition of ANY KIND can alter the Mueller Matrix results I believe that a validation experiment is needed.

The referred question #8: Figure 6, was any validation conducted on the citoplasm composition?

Reply:

Thank you for expressing your concerns, we appreciate your insights for improving the manuscript. The primary objective of this manuscript is to propose a method which is capable of decomposing the biological tissue into a basis of microstructural clusters, and in particular, to show that cluster 3 is potentially meaningful for differentiating HCC and ICC. H&E images are the gold standard in pathology. In this work, several experienced pathologists examined and evaluated the projection results on the H&E images, and infer that the spatial distribution of cluster 3 and 5 labels likely correspond to cytoplasm and collagen. We believe that cluster 3 represents a type of cytoplasm with a specific polarization nature, and its proportion potentially differentiates ICC and HCC. To validate and strengthen this claim, we provide additional projection results on high-definition H&E stained images as supplementary documents (supplementary fig.3), as well as on H&E images under 40X magnification (supplementary fig.4).

Studying which type of microstructural variation corresponds with particular polarization signature will require other molecular specific staining or modalities microscopy such as SHG or super resolution imaging, which is beyond the scope of this paper.

We modified the manuscript in the Results and Discussion section to further improve clarity.

Results section:

“ We now study the differences in tissue microstructural composition between HCC and ICC pathological samples, in an attempt to distinguish them. Likewise, we can first visualize the variation of tissue microstructural composition by observing the animated heatmap (provided in *Supplementary video 2*) that samples gradually from HCC to ICC tissues, which provides clear visualization of the subtle difference in local density. The microstructures characterized by cluster 5, i.e., collagen fiber, is abundant in ICC samples, but not in HCC samples, as seen in Fig.6a. In contrast, cluster 3, a subtype of cytoplasm, is abundant in HCC sample while not in ICC samples. Cluster 5 is the main focus as the polarization marker for HCC and ICC distinction. For comparison, Fig.6b shows the box-whisker plot of the cluster 5 area proportion for both HCC and ICC samples. It is observed that most of HCC samples' cluster 5 proportion ratio is less than 10%, while that of the

majority of ICC samples are above 10%. For discriminating ICC from HCC samples, the area proportion of cluster 5 yields an AUC of 84.94%. We also experimented with cluster 3, using it as the polarization marker yields an AUC of 71.69%, as seen in Fig.6c. This implies that to a certain degree, the cytoplasm composition in HCC and ICC are different, *and such difference can be reflected in the proportion of cluster 3, a polarization subtype of cytoplasm*. It is noted that in this work identifications between the six polarization pixel clusters and their corresponding pathological composition were made by *experienced* pathologists based on color images of the H&E-stained slides, which show the boundaries of cells. *Supplementary fig.3 contains additional projection results onto high definition H&E images, where the labelled region of cluster 3 is highlighted with brown color, similar to the color scheme of IHC. Supplementary fig.4 shows the projection results under even higher magnification (40X)*. Such identification between polarization and microstructural features will be further improved using other molecular specific staining methods or super-resolution techniques. *While the current results are based on a relatively limited samples size, we aim to establish a correspondence between polarization features and microstructural variations in cytoplasm in future works.* ”

Discussion section:

“We demonstrate how polarization markers can be extracted and potentially utilized for pathological applications. Tissue microstructural composition analysis enables visualization of tissue structure transition in polarization space during pathological variation. It is shown that cluster 2 correlates strongly with tumor. Evidence shows that cluster 2 can serve as a polarization marker to quantitatively characterize HCC samples with different differentiation degrees and separate HCC from normal liver tissues with an AUC of 95%. Differentiation of ICC from HCC is a known challenging pathological task. In this study, we identify a polarization marker to distinguish ICC from HCC on the H&E-stained slides, achieving an AUC of 84.94%. Through the transformation from polarization feature space to pathological feature space, it informs pathologists that cytoplasm and collagen contribute the most toward classification, which is consistent with the findings of medicine ⁴⁶. *In particular, we observe that cluster 3 is a polarization subtype of cytoplasm that is potentially meaningful for differentiating HCC and ICC, and the correspondence between polarization features and microstructural variations in cytoplasm may be further studied using molecular staining or super-resolution imaging methods*. Therefore, this method may supplement the immunostaining analysis, providing interpretable and quantitative polarization markers for pathological diagnosis and prognosis prediction, and even reveal “invisible” super-resolution features to pathologists in an intuitive and visualized manner. ”

Supplementary Fig 3. Additional cluster 3 projection results on H&E images from both HCC and ICC ROIs, highlighted areas are the labelled areas of cluster 3, represented by a brown-red tint. Cluster 3 appears to be related to cytoplasm structures.

Supplementary Fig 4. Cluster 3 projection results on H&E images under 40X magnification, highlighted areas are the labelled areas of cluster 3, represented by a brown-red tint.

REVIEWERS' COMMENTS:

Reviewer #3 (Remarks to the Author):

The authors have sufficiently replied to my questions, although their claims still seem to be exaggerated.